# Subducted carbon weakens the forearc mantle wedge in a warm subduction zone

Ryosuke Oyanagi [1,2] ✉ & Atsushi Okamoto [3] ✉

Subducting oceanic plates carry large amounts of carbon into the Earth's interior. The subducted carbon is mobilized by fluid and encounters ultramafic rocks in the mantle wedge, resulting in changes to the mineral assemblage and mechanical properties of the mantle. Here, we use thermodynamic modeling of interactions between carbon-bearing multi-component fluids and mantle rocks to investigate the down-dip variation in mineral assemblage in the forearc mantle along subduction megathrusts. We found that fluids rich in aqueous carbon are preferentially generated in a warm subduction zone (e.g., Nankai, SW Japan), causing a change in mineral assemblage from serpentine-rich at the mantle wedge corner to talc + carbonate-rich at greater depths. The transition caused by the infiltration of aqueous carbon may influence the depth of the boundary between the seismogenic and aseismic zones, and the down-dip limit of episodic tremor and slip.

Earth is not only a planet of water but also of carbon, which has given rise to the evolution of carbon-based life[1–4]. The geological carbon cycle, of which subduction zones are a key component, regulates the habitability of Earth. However, the fate and consequences of subducted carbon and the extent of carbon cycling at the forearc mantle remain poorly understood[3].

Subducted carbon, which is hosted mainly in sediment[2,3,5], is partly mobilized by fluids released from the subducting slab and transported into the overlying forearc mantle wedge[6–8], resulting in a change in the mineral assemblage of the wedge above the plate interface (Fig. 1a)[9–13]. Geological observations suggest that mass transfer of aqueous carbon and dissolved major elements (e.g., $SiO_2$ and $Al_2O_3$) occurs across lithological boundaries such as the slab–mantle interface[10,14–16]. The mass transfer of the dissolved major elements results in the formation of mechanically weak minerals (i.e., talc and chlorite) at the base of the forearc mantle wedge, thereby controlling the occurrence of episodic tremor and slip (ETS) in the region of the mantle wedge corner (MWC) in a warm subduction zone[17–22] and slab–mantle decoupling in deeper parts of the subduction zone[23–25]. Therefore, the distribution of weak minerals and mineralogical variations along the subducting megathrust may be important in understanding the transition from the seismogenic to aseismic zones.

Fluids derived from the subducted sediments are regarded as one of the dominant fluid sources of fluid–rock interactions in the forearc mantle throughout infancy to maturity of the subduction zone[26]. The accurate prediction of the products of fluid–rock interactions requires knowledge of the chemical composition of the infiltrating fluid in a multi-component system. Based on petrological thermodynamic models, the interaction of slab-derived fluid with overlying forearc mantle peridotite has been studied for pure $H_2O$[23,26–28] or $H_2O$-rich fluids that contain dissolved rock components, but without considering aqueous carbon[29–32]. In contrast, the interaction of rock with fluid containing aqueous carbon and dissolved rock components has been investigated for specific $P–T$ conditions[10] or several sets of $P–T$ conditions along a subduction geotherm[33]. However, thermal structures along the slab–mantle interface (Fig. 1b, c)[24,34] and the chemical compositions of subducting sediment[35] (including organic and inorganic carbon species; Fig. 1d)[36] are variable in subduction zones. Consequently, the fluid composition and fluid flux into the mantle wedge vary with depth and in different arcs[7,27,28,37,38], leading to corresponding variations in the mineral assemblage of the mantle wedge. A coupled time-evolving subduction zone thermal model and phase equilibria show that temperature evolution from subduction infancy to maturity results in different degrees of serpentinization of

[1]School of Engineering and Science, Kokushikan University, Tokyo 154-8515, Japan. [2]Research Institute for Marine Geodynamics (IMG), Japan Agency for Marine-Earth Science and Technology (JAMSTEC), Yokosuka 237-0061, Japan. [3]Department of Environmental Studies for Advanced Society, Graduate School of Environmental Studies, Tohoku University, Sendai 980-8579, Japan. ✉e-mail: oyanagir@kokushikan.ac.jp; atsushi.okamoto.d4@tohoku.ac.jp

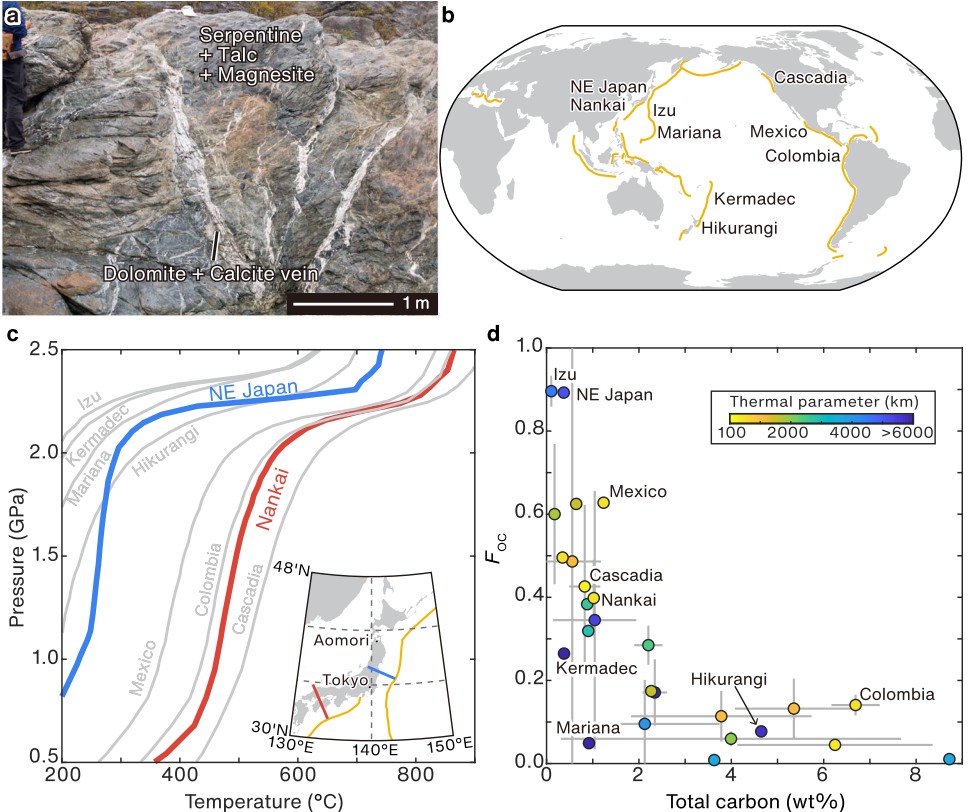

**Fig. 1 | Variations in subducting carbon among subduction zones and the thermal structure of subduction zones. a** Photograph of a paleo-mantle wedge in a subduction zone (Higuchi serpentinites, Japan), showing a talc + carbonate-rich assemblage as a consequence of fluid infiltration into the hydrated mantle[10]. **b** Global map of subduction zones. The yellow lines indicate subduction trenches[75]. **c** Slab-top thermal structure in northeastern Japan and Nankai subduction zones[24], along the transects indicated in the inset. The slab-top geotherms of other subduction zones[24] listed in (**b**) are shown by light gray lines. **d** Variations in carbon contents in subducting sediments. $F_{OC}$ represents the mass fraction of organic carbon relative to the total carbon in the sediment. A low $F_{OC}$ indicates the subducted carbon is dominated by inorganic carbon (carbonate), whereas a high $F_{OC}$ indicates the subducted carbon is dominated by organic carbon (e.g., carbonaceous material). Each symbol is colored according to a thermal parameter[36], which is an indicator of whether a subduction zone is warm (low value) or cold (high value). If available, error bars represent one standard deviation.

the forearc mantle wedge[26]. Understanding on the nature of depth-related variations in the mineral assemblage of the forearc mantle and its dependence on the thermal structure of the subduction zone is still limited, especially with respect to the interaction with multi-component fluids.

In this study, we conducted thermodynamic modeling of interactions between carbon-bearing fluids and mantle rocks in subduction zones. Minimization of the Gibbs free energy of mineral solid solutions and electrolyte speciation within a pseudo-open system[39] was used to constrain chemical mass transfer during fluid–rock interaction. In the modeling, all of the major rock components (e.g., C, $SiO_2$, and $Al_2O_3$) were considered to be soluble in the fluid, to be consistent with geologically observed mass transfer across the slab–mantle interface[14–16]. Moreover, we considered the composition of the subducting carbon-bearing sediments that differ by each subduction zone (Fig. 1b, d)[35,36]. Our calculation results constrain the composition of the fluid transported from the subducting sediments to the mantle wedge, and the changes in mineral assemblage at the base of the mantle wedge along the subduction megathrust. The calculated changes in the mineral assemblage are used to discuss their relationship with the transition between the seismogenic and aseismic zones.

## Results and discussion

Thermodynamic calculations were conducted for two contrasting subduction megathrusts in northeastern Japan and Nankai (southwestern Japan; Fig. 1b, c). These regions are considered to be representative of cold and warm subduction zones, respectively[37,40]. We used the slab-top geotherms calculated by Wada and Wang[24] because they are geotherms of specific areas that cross the boundary between the seismogenic and aseismic zones, rather than using the average geotherm over an area of hundreds of kilometers in scale. Northeastern Japan is a typical cold subduction zone with a slab-top geotherm of 229 °C at 1.0 GPa and 292 °C at 2.0 GPa (-60 °C GPa$^{-1}$), whereas Nankai is a warm subduction zone with a slab-top geotherm of 468 °C at 1.0 GPa and 570 °C at 2.0 GPa (-100 °C GPa$^{-1}$; Fig. 1c)[24]. The subducting sediments in the northeastern Japan subduction zone contain 0.38 wt% total carbon (TC) and are dominated by organic carbon (OC) with a $F_{OC}$ (i.e., the mass fraction of OC relative to TC) value of 0.89 (Fig. 1d; Supplementary Table 1)[36]. In the Nankai subduction zone, the subducting sediments contain 1.03 wt% TC as both OC and inorganic carbon ($F_{OC}$ = 0.41; Fig. 1d; Supplementary Table 1)[36].

### Predicted fluid compositions and fluid flux

Figure 2a shows the predicted compositions of fluid in equilibrium with the subducting sediments in the northeastern Japan subduction zone. The fluid is rich in Na (0.13–0.88 mol kg$^{-1}$) and Si (0.04–2.7 mol kg$^{-1}$) at 25–90 km depth, and the concentrations of these elements increase with depth (i.e., increasing pressure and temperature). The C concentration decreases gradually with increasing depth, from 0.06 mol kg$^{-1}$ at 25 km depth to 0.02 mol kg$^{-1}$ at 70 km depth, and exhibits an abrupt increase to 0.91 mol kg$^{-1}$ at 76 km depth

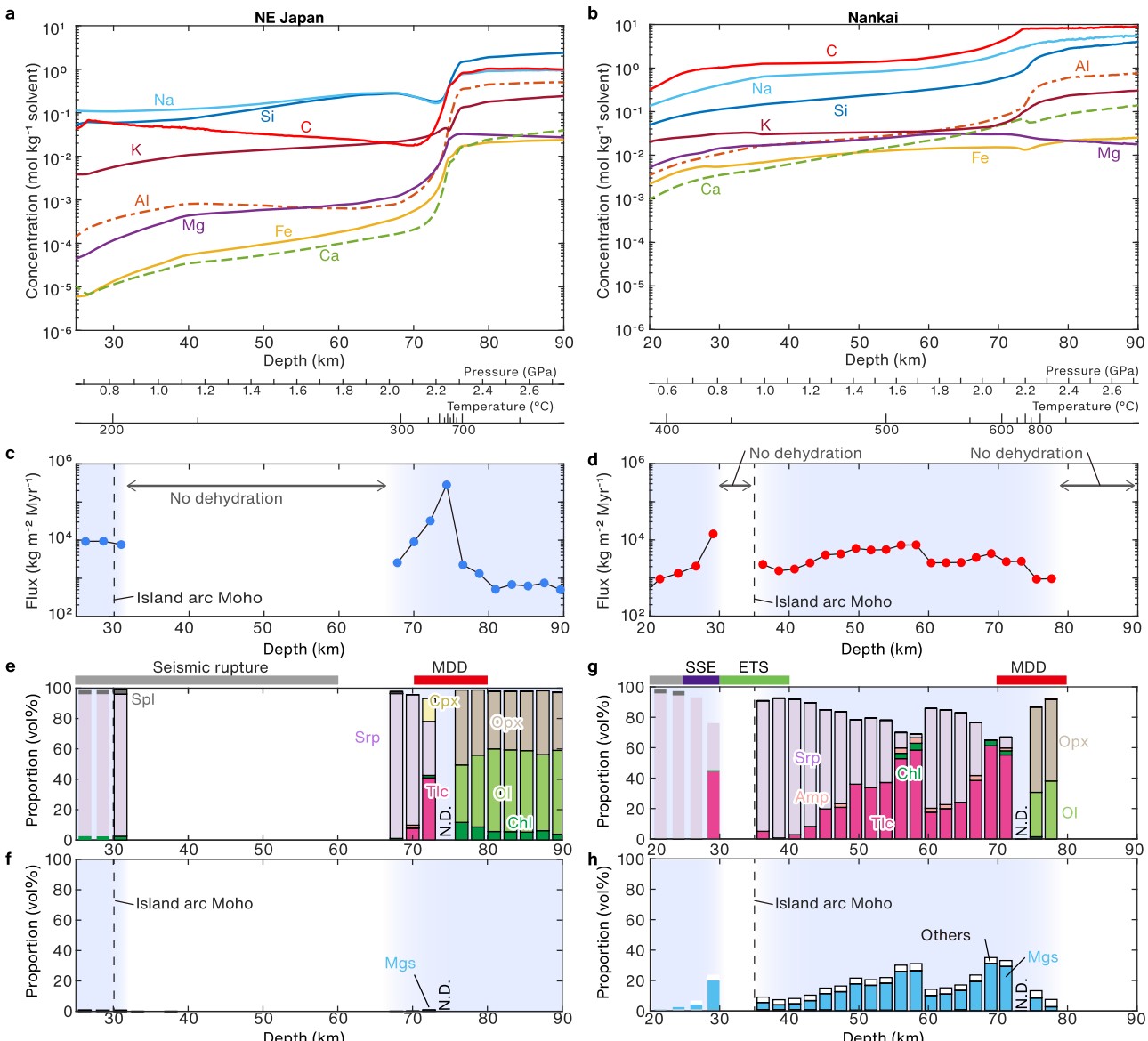

**Fig. 2 | Results of thermodynamic calculations. a–b** Predicted composition of fluid in equilibrium with metasedimentary rocks along the *P–T* path of **a** northeastern Japan and **b** Nankai subduction zones. **c–d** Dehydration flux of subducting sediments in the **c** northeastern Japan and **d** Nankai subduction zones. The blue shading indicates the depths of predicted dehydration of the subducting slab. The vertical dotted line represents the Moho beneath an island arc[24]. **e–f** Predicted mineral proportions (**e**) and proportion of carbonate (**f**) in the mantle wedge adjacent to subducting sediments for *t*/*L* =1 (Myr m⁻¹) in the northeastern Japan subduction zone. **g–h** Predicted mineral proportions (**g**) and proportion of carbonate (**h**) in the mantle wedge adjacent to subducting sediments for *t*/*L* =1 (Myr m⁻¹) in the Nankai subduction zone. In (**e–h**), mineral proportions in regions shallower than the depth of the island arc Moho are shown in light colors for reference. MDD maximum depth of decoupling[24], SSE slow slip event[17,18], ETS episodic tremor and slip[17,18]. N.D. (not determined) indicates the mineral proportions were not calculated due to numerical instability. The blue and white backgrounds indicate the depths at which dehydration was predicted and not predicted, respectively. Srp serpentine, Tlc talc, Chl chlorite, Mgs magnesite, Dol dolomite, Qz quartz.

(Fig. 2a). By contrast, in the Nankai subduction zone, carbon is the element with the highest concentration in the fluids, ranging from 0.12 mol kg⁻¹ at 20 km depth to 8.0 mol kg⁻¹ at 72 km depth (Fig. 2b). The C concentrations are relatively constant at 72–90 km depth (7.9–8.4 mol kg⁻¹; Fig. 2b). The concentrations of elements in the fluids in equilibrium with the metasedimentary rocks show an increase with depth (Fig. 2b). The Na and Si concentrations in the fluids are higher than the concentrations of other elements, but are approximately one order of magnitude lower than the carbon concentrations. The Si concentrations range from 0.04 mol kg⁻¹ at 20 km depth to 5.1 mol kg⁻¹ at 90 km depth (Fig. 2b).

In the northeastern Japan subduction zone, our model predictions show that dehydration of subducting sediments occurs at depths of 25–30 and 68–90 km (Fig. 2c), which correspond to the breakdown of stilpnomelane and lawsonite + amphibole, respectively (Supplementary Discussion 1). Dehydration is not predicted to occur at depths of 32–68 km (Fig. 2c). The predicted fluid fluxes are 2.1–9.4 × 10³ kg m⁻² Myr⁻¹ at 23–30 km depth, increase from 2.5 × 10³ kg m⁻² Myr⁻¹ at 68 km depth to 282.2 × 10³ kg m⁻² Myr⁻¹ at 75 km depth, and subsequently decrease to 0.5 × 10³ kg m⁻² Myr⁻¹ at 90 km depth (Fig. 2c). In contrast, in the Nankai subduction zone the model predictions indicate the dehydration of subducting sediments at depths of 20–30 km and 35–78 km (Fig. 2d) due to the breakdown of chlorite (Supplementary Discussion 1). The predicted fluid flux from the sediments shows a peak at 30 km depth (1.4 × 10⁴ kg m⁻² Myr⁻¹), and a flux of 0.1–0.7 × 10³ kg m⁻² Myr⁻¹ is predicted for the region below the Moho (i.e., below 35 km depth; Fig. 2d).

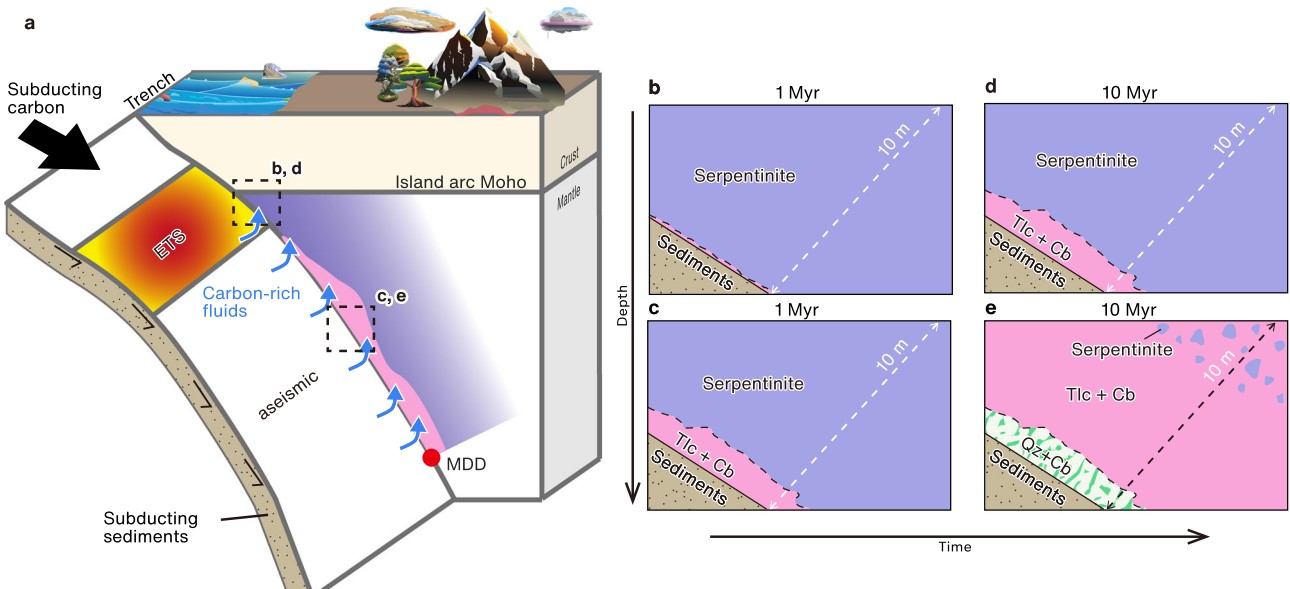

**Fig. 3 | Schematic illustration of down-dip variations in slip behavior and the mineral assemblage of the mantle wedge in the Nankai subduction zone.**
**a** Cross-sectional illustration of the Nankai subduction zone. Episodic tremor and slip (ETS) occur at 30–40 km depth around the mantle wedge corner (MWC)[17,18], whereas greater depths are aseismic. **b**, **c** Predicted mineral distribution (after 1 Myr of fluid activity) along the subduction megathrust around **b** the MWC and **c** at greater depths (50–70 km), as indicated by the black dotted square in (**a**). **d**, **e** Predicted mineral distribution (after 10 Myr of fluid activity) along the subduction megathrust around **d** the MWC and **e** at greater depths (50–70 km), as indicated by the black dotted square in (**a**). Tlc talc, Cb carbonate, Qz quartz.

## Predicted mineral assemblages of the forearc mantle

Using the predicted fluid compositions and fluid fluxes at various depths in the subduction zones, we calculated the mineral assemblage at the base of the mantle wedge in contact with sediment during fluid infiltration into the hydrated mantle wedge ("Methods"). Our model results were controlled by $t/L$ (Myr m$^{-1}$), where $t$ (Myr) is the timescale of fluid infiltration and $L$ (m) is the length or thickness of the forearc mantle measured normal to the dip of the slab.

For the northeastern Japan subduction zone and $t/L = 1$ (Myr m$^{-1}$), hydrous minerals and carbonate minerals are predicted to be absent at depths of 30–68 km (Fig. 2e, f), where dehydration is not predicted to occur (Fig. 2c). At depths of 68–90 km, our calculations indicate that dehydration of subducting sediments leads to fluid infiltration into the mantle wedge, resulting in the formation of talc (10 vol%) at a depth of 70 km and chlorite (10 vol%) at depths of 78–90 km, whereas the occurrence of carbonate minerals is limited (Fig. 2e, f). In the case of $t/L = 10$ (Myr m$^{-1}$), talc + quartz-rich rock and chlorite + orthopyroxene-rich rock are predicted at depths of 68–80 km and 80–90 km, respectively (Supplementary Discussion 2).

The duration of subduction (i.e., the age of the arc) in the northeastern Japan subduction zone is 46–60 Myr[41]. Assuming that fluid supply from the subducting sediments continues for 50 Myr, the model results in the case of $t/L = 1$ Myr m$^{-1}$ (Fig. 2e, f) can be regarded to reflect the mineral assemblage at the base of the forearc mantle in a layer that is 50 m thick. The calculations for the northeastern Japan subduction zone suggest the limited occurrence of hydrous minerals (serpentine and talc) and carbonate minerals (Fig. 2e), due to the lack of fluid infiltration from subducting sediments into the mantle wedge at depths of 32–68 km (Fig. 2c). Our calculations with $t/L = 1$ (Myr m$^{-1}$) suggest the local occurrence of a talc-bearing layer without carbonate minerals at a depth of 70 km (Fig. 2e), where sediment dehydration is predicted to occur (Fig. 2c).

For the Nankai subduction zone, where the dehydration of sediments is predicted at depths of 35–78 km (i.e., below the Moho, which is located at 35 km depth; Fig. 2d), the calculations with $t/L = 1$ (Myr m$^{-1}$) indicate the occurrence of talc (1–60 vol%) and carbonate (5–32 vol%) as a consequence of fluid infiltration (Fig. 2g, h). The predicted modal abundances of talc and carbonate minerals increase from 35–40 km depth (talc = 0.6–5.0 vol%; carbonate minerals = 4.0–5.5 vol%) to 40–50 km depth (talc = 2.8–36.0 vol%; carbonate minerals = 4.6–18.8 vol%) (Fig. 2g, h). In the case of $t/L = 10$ (Myr m$^{-1}$), talc + carbonate (magnesite) occur around the MWC (35–50 km depth), but the proportion of talc decreases with depth, with a mineral assemblage of quartz + carbonate (magnesite) at greater depths (50–74 km; Supplementary Discussion 2).

The tectonic history of the Nankai subduction zone is complex, involving transform motion between the Pacific and Eurasia plates, the cessation of Pacific plate subduction, and subsequent subduction of the younger Philippine Sea plate since 17 Ma[42]. Based on this geological history, we assume fluid infiltration for 1–10 Myr in the Nankai subduction zone. Our calculation suggests that after fluid infiltration for 1 Myr, a 1-m-thick layer of talc-poor serpentinite is formed at the base of the forearc mantle around the MWC, with a 1-m-thick layer of talc + carbonate-rich rock at greater depths (based on the results for $t/L = 1$ Myr m$^{-1}$; Fig. 2g, h and 3a–c). After fluid infiltration for 10 Myr, a 10-m-thick basement layer of talc-poor serpentinite and talc + carbonate-rich rock is predicted to be formed around the MWC and at greater depths, respectively (based on the results for $t/L = 1$ Myr m$^{-1}$; Fig. 2g, h and 3d, e). Moreover, at fluid infiltration for 10 Myr, the 1-m-thick basement layers of talc + carbonate-rich rock and quartz + magnesite-rich rock are also predicted at a depth of 35–50 km and 50–74 km depth, respectively (based on the results for $t/L = 10$ Myr m$^{-1}$; Supplementary Discussion 2; Fig. 3d, e).

Overall, our modeling approach shows contrasting predictions of the mineral assemblage at the base of the mantle wedge between northeastern Japan and the Nankai subduction zone. The model results show little change in the case that interaction between fluids derived from subducting sediments and anhydrous mantle (Supplementary Discussion 3) is considered. Moreover, the dehydration of subducting altered oceanic crust (Supplementary Discussion 4) and underplating of subducting sediments (Supplementary Discussion 5) influence the fluid flux into the mantle wedge and result in a marked difference in the proportion of talc between the region around the MWC and greater depths.

## Controls on the mineral assemblage of the forearc mantle

In the forearc mantle, talc can be formed in serpentinite as a result of the infiltration of aqueous silica-rich fluids[31,43]:

$$\underset{\text{serpentine (antigorite)}}{Mg_{48}Si_{34}O_{85}(OH)_{62}} + 2\,SiO_{2(aq)} = 16\,\underset{\text{talc}}{Mg_3Si_4O_{10}(OH)_2} + 15\,H_2O, \quad (1)$$

or aqueous carbon-rich fluids[44]:

$$2\,\underset{\text{serpentine (antigorite)}}{Mg_{48}Si_{34}O_{85}(OH)_{62}} + 45\,CO_{2(aq)} = 17\,\underset{\text{talc}}{Mg_3Si_4O_{10}(OH)_2} + 45\,\underset{\text{magnesite}}{MgCO_3} + 45\,H_2O.$$

$$(2)$$

In Eqs. (1) and (2), the serpentine composition is represented by antigorite. Although the cold geotherm of the northeastern Japan subduction zone might result in the stability of lizardite or chrysotile rather than antigorite, the reactions would be similar to those shown. The prediction of talc formation (without carbonate minerals) at 70 km depth in the northeastern Japan subduction zone (Fig. 2e) would be via Eq. (1), as the predicted fluid has low concentrations of C and high concentrations of Si (Fig. 2a). In contrast, in the Nankai subduction zone, a talc + carbonate assemblage is predicted at depths of 35–72 km, and the proportions of both minerals increase with depth at a similar rate (Fig. 2g, h), indicating that talc is mainly formed by the infiltration of carbon-rich fluid (Eq. 2) rather than silica-rich fluid (Eq. 1). The significant effect of carbon-rich fluid on talc formation in the Nankai subduction zone is supported by additional results that show (1) the predicted fluids have high concentrations of C as compared with Si (Fig. 2b); (2) the solubility of carbon increases under high-temperature and low-pressure conditions[45]; and (3) in carbon-free systems, two to four times more fluid is required to form the same amount of talc in the mantle wedge compared with carbon-bearing systems (Supplementary Discussion 6). As such, aqueous carbon in fluid that originates from the carbon in subducting sediments is a primary contributor to talc formation in a warm subduction zone.

The thermal structures, amounts of subducted carbon, and dominant form of carbon (i.e., organic or inorganic carbon) vary significantly amongst subduction zones worldwide (Fig. 1c, d). To elucidate the key factors that control talc and carbonate mineral formation, we conducted parameter studies to investigate the relative sensitivities of TC and $F_{OC}$ using the average marine sediment composition (i.e., global subducting sediment, GLOSS)[35] and GLOSS with modified TC contents. The efficiency of talc formation was evaluated using the parameter $\xi_{Tlc}$ (mol fluids kg$^{-1}$ rock), which represents the amount of fluid required for the formation of talc. The results (Supplementary Discussion 7) show that (1) $\xi_{Tlc}$ is lower in the Nankai subduction zone than in the northeastern Japan subduction zone, despite the variations in $F_{OC}$ and TC; and (2) low $F_{OC}$ tends to reduce $\xi_{Tlc}$ in both subduction zones.

Additional parameter studies were conducted using five types of subducting sediment (carbonate sediment, chert, pelagic clay, terrigenous sediment, and turbidite). The results (Supplementary Discussion 8) show that fluids derived from the dehydration of carbonate sediment and chert do not result in talc formation in the forearc mantle wedge because these rocks, dominated by carbonate minerals or quartz, provide minimal $H_2O$ into the forearc mantle wedge. The results also show that $\xi_{Tlc}$ increases in the order of pelagic clay, turbidite, and terrigenous sediment. These results suggest that the subduction geotherm and sediment type are the main controls on the occurrence of talc + carbonate in the mantle wedge.

## Changes in mineral assemblage in other subduction zones

We test whether our results for the Nankai subduction zone can be generalized more broadly to warm subduction zones by applying our methodology to the Cascadia subduction zone. Calculations for the Cascadia subduction zone (Supplementary Discussion 9) show that infiltration of fluids derived from the subducting sediment (mainly turbidite[46] and low $F_{OC}$) into the mantle wedge also results in the formation of a talc + carbonate assemblage in the forearc mantle at the Cascadia subduction zone. Moreover, the predicted talc fraction increases with depth, as predicted in the modeling of the Nankai subduction zone. The talc fraction increases from 0% at the MWC (35 km depth) to 60% at 50 km depth in the Cascadia subduction zone.

The results of the typical cold and warm subduction zones (Fig. 2) and parameter studies on sediment compositions (Supplementary Discussions 6–8) can be used to infer the presence of talc + carbonate at the base of the mantle wedge in other subduction zones. Parameter studies of cold subduction zones suggest that TC and $F_{OC}$ affect talc formation. The value of $\xi_{Tlc}$ decreases with increasing TC in sediments in northeastern Japan, whereas this trend is weaker in the Nankai subduction zone (Supplementary Discussion 3). In the northern Hikurangi cold subduction zone, where subducting sediments have high TC (4.6 wt%) and low $F_{OC}$ (0.08; Fig. 1b, c)[36], a talc + carbonate-rich rock is expected to form if fluid infiltrates the mantle wedge.

The Mexico subduction zone is a warm subduction zone similar to the Nankai and the Cascadia subduction zone (Fig. 1b, c). Subducting sediment in the Mexico subduction zone is dominated by pelagic clay[46], and carbon occurs mainly in carbonate minerals (low $F_{OC}$; Fig. 1b, d)[36]. Previous thermodynamic modeling of the Mexico subduction zone using a carbon-free system[30] has estimated the amounts of talc formed by Si metasomatism (Eq. (1)). Our calculations suggest that the amount of talc predicted in this system is greater if subducted carbon in the Mexico subduction zone (carbonaceous materials and carbonate minerals)[36] is considered. In contrast, Colombia subduction zone (Fig. 1b, c) is an exception among warm subduction zones, where talc is unlikely to occur because the subducting sediments are solely carbonate sediments[46] and provide minimal $H_2O$ to the forearc mantle wedge[28] (Supplementary Discussion 8).

## Comparison with seismic observations

In the northeastern Japan subduction zone, the seismic properties of the forearc mantle at depths of 30–60 km are interpreted to reflect a low proportion of serpentine[47], whereas the low-velocity layer at 60–80 km depth[47] suggests the presence of hydrous minerals[48]. These observations are consistent with our calculations of the depth of dehydration of subducting sediment (Fig. 2c). In contrast, in the Nankai subduction zone, the low-velocity layer ($V_P = 7.0$–7.5 km s$^{-1}$ and $V_S = 3.8$–4.1 km s$^{-1}$) at the MWC has been interpreted as serpentinized mantle[49,50]. However, the seismic velocities calculated from mineralogy (Supplementary Discussion 10) suggest that it is difficult to distinguish between the serpentinite and talc + carbonate-rich assemblages based solely on seismic observations, as suggested previously[51]. Therefore, the low-velocity layer in the Nankai subduction zone could also be due to a talc + carbonate-rich layer at the base of the mantle wedge.

## Rheological consequences of heterogeneous talc formation along a subduction megathrust

The calculated results for northeastern Japan (Fig. 2e) suggest that almost all the carbon in subducting sediments (0.44 Mt C yr$^{-1}$)[36] is carried to depths of ~90 km. In contrast, our calculation shows that some of the carbon in the subducted Nankai sediments is dissolved in fluids and mobilized into the mantle wedge at depths of <80 km (Fig. 2f). We have estimated the rate of carbon uptake in the mantle wedge in the Nankai subduction zone (Method) to be 0.005 Mt C yr$^{-1}$, which is only 0.5% of the subducting carbon in the Nankai subduction zone (1.00 Mt C yr$^{-1}$)[36]. Accordingly, in both subduction zones, little of the carbon in subducted sediments is recycled into the mantle wedge, and the carbon uptake in the mantle wedge may not affect the current

model of global carbon cycling and balance between carbon influx and outflux[3]. Nevertheless, the subducted carbon can have significant effects on the rheology of the plate interface, as discussed below.

Geophysical observations have revealed that the subducting slab becomes fully coupled to the overlying mantle wedge below 70–80 km depth[24]. Previous closed-system calculations suggest the maximum depth of decoupling (MDD) could be controlled by the stability of talc[23], as this is the mineral with the lowest frictional coefficient amongst mantle minerals[52,53]. In the northeastern Japan subduction zone, our open-system calculations suggest talc disappearance at a depth of 70–75 km, synchronous with the appearance of chlorite (Fig. 2e). Chlorite is a weak mineral that accommodates stable slip[54]. Because of the low proportions of chlorite and talc (<10 vol%; Fig. 2e), the predicted occurrence of these minerals at 70–90 km depth may not weaken the rocks, leading to coupling between the slab and mantle. In contrast, our calculations for the Nankai subduction zone show talc disappearance at depths of >76 km (Fig. 2f). Talc stability in our open-system calculation is similar to that in the closed-system calculation[24], suggesting that talc disappearance at 70–80 km depth controls the extent of slab–mantle coupling in the Nankai subduction zone.

The predicted variation in talc distribution at the slab–mantle interface in different subduction zones has a significant effect on seismic activity. In northeastern Japan, the hypocenters of inter-plate earthquakes are widely distributed at depths greater than the Moho, reaching depths of 50–60 km[17]. These observations, as well as the seismic constraints, are consistent with our calculations, which reveal a lack of sediment dehydration at depths of 30–68 km (Fig. 2c) and the presence of hydrous minerals at depths of 68–90 km (Fig. 2e).

In the Nankai subduction zone, the down-dip limit of the seismogenic zone is located at a depth of 20 km. Long-term slow slip events (SSEs; 25–30 km depth) and ETSs (30–40 km depth) have been observed, and greater depths (>40 km) are aseismic (Fig. 3)[17,18]. Our calculations show that talc + carbonate rocks are widely distributed from the shallow MWC (35 km) to the MDD (70–80 km)[24] in the Nankai subduction zone (Fig. 2g, h). However, our calculations suggest the amount of talc is lower at the MWC (35–40 km depth) and increases significantly at depths of 40–70 km (Fig. 2g, h), in response to the sediment-derived fluid containing high concentrations of aqueous carbon (Supplementary Discussion 2). The frictional properties of talc control the occurrence of deformation even at high pressures and temperatures[55], and it exhibits low frictional strength and pronounced velocity-strengthening behavior[56] that induces aseismic (stable) sliding of the subduction zone megathrust[57]. Moreover, experiments with talc–calcite[58] and serpentine–talc[52,53] mixtures suggest that talc controls the overall deformation and frictional behavior of rocks when the amount of talc is increased to 10–20 vol%. The formation of a layer of carbonate + quartz-rich rock was predicted at depths of 50–74 km (Supplementary Discussion 2; Fig. 3e); however, the predicted thickness of the carbonate + quartz-rich rock at the interface is less than 10% of that of talc + carbonate rock within the mantle wedge at the same depth (Fig. 3e). Consequently, the talc-rich base of the mantle wedge at greater depths (i.e., 50–70 km) would be rheologically weak compared with the talc-poor base of the mantle wedge at MWC, and the increasing proportion of talc in serpentinite with depth (Fig. 2g) could facilitate the transition from the ETS (seismic) zone to the aseismic zone. The present study did not explore the specific mechanism underlying ETS rupture; however, the results suggest that the talc-rich assemblage formed by the infiltration of carbon-rich fluids regulates the down-dip limit of the region of ETS in warm subduction zones.

In summary, we conclude that the infiltration of carbon-bearing fluids leads to spatially heterogeneous mineralogy in the mantle wedge and potentially influences the mechanical properties of the slab–mantle interface, despite having little effect on the global carbon cycle. The rheology of the slab–mantle interface and the occurrence of earthquakes are influenced by the carbon cycle from the surface to the deep Earth, as well as the thermal structure of subduction zones.

## Methods

### Overview
The thermodynamic calculations were performed in two steps. First, the chemical composition and flux of the fluids released from the subducting sediments were calculated. These data were then used to predict the mineral assemblage of the forearc mantle wedge along the subduction megathrust.

### Thermodynamic calculations of the compositions of fluids released from subducting sediment
Thermodynamic calculations for the fluids were conducted using the Deep Earth Water (DEW) model[59] in Perple_X[60] version 7.1.2. We used the lagged speciation algorithm of Perple_X to calculate the thermodynamic equilibrium between solid and solvent[39]. We used the DEW19HP622ver_elements.dat dataset, which is based on the TC-DS622 version of the mineral database[61,62], with aqueous species from the DEW model 2019[63]. The composition of the fluid solvent was adaptively determined, and the concentrations of molecular volatiles were used as the solvent species if their molar fraction was >$10^{-5}$. When molecular volatiles were included as the solvent species, their aqueous species were excluded from the calculations. High molecular weight organic compounds (e.g., $HCOO^-$ and $CH_3COO^-$) were excluded from the computations. $Mg(SiO_2)(HCO_3)^+$ and $H_2CO_{3(aq)}$ species were also excluded because they occurred in unrealistically high concentrations[64,65] (for details, see the input files in the "Data availability" section). The equations of state for $H_2O$ and $CO_2$ are the Pitzer and Sterner equations of state (PSEoS)[66], and the equation of state for $CH_4$ is a modified Redlich–Kwong (MRK) equation[67]. The DQF parameter of the ferric chlorite end-member ("f3clin" in DEW19HP622ver_elements.dat) was set to 40,000 J mol$^{-1}$ [68]. The solid-solution models used for the analysis are summarized in Supplementary Table 2.

Modeling of the mineral assemblages in the sediments was conducted in the Si–Al–Fe–Mn–Mg–Ca–Na–K–$H_2$–C–$O_2$ system. Thermodynamic calculations were conducted at the $P$–$T$ trajectories along the slab-top geotherms of the Nankai and northeastern Japan subduction zones[24] using the whole-rock compositions of subducting sediments observed at each subduction zone (Supplementary Table 1). The major element compositions of the subducting sediments[35] and the organic and inorganic carbon contents in the sediments in each subduction zone[36] were used for the modeling. The $Fe^{3+}/\Sigma Fe$ ratio was set to 0.23, based on the global average for metapelites[69]. The mineral assemblage, volume proportion of minerals (vol%), amounts of water (wt%) and carbon (wt%) in the rocks, density of the sediments, and fluid compositions were extracted using the sub-program WERAMI in Perple_X. The results are presented in Supplementary Discussion 1.

### Calculating the fluid flux from subducting sediment
Based on mass balance considerations, the fluid flux from the subducting sediments ($J$; kg fluid m$^{-2}$ Myr$^{-1}$) was calculated as follows:

$$J = \rho_{sed} \frac{hv}{l} \triangle m_{fluid}, \tag{3}$$

where $\rho_{sed}$ (kg m$^{-3}$) is the density of the subducting sediments, $h$ (m) is the thickness of the subducting sediments, $v$ (m Myr$^{-1}$) is the subduction velocity, and $l$ (m) is the length along the slab dip. $\Delta m_{fluid}$ is the change in the fluid mass concentration (kg fluid kg$^{-1}$ rock) over interval $l$. The $v$ values were set to 100,000 m Myr$^{-1}$ and 40,000 m Myr$^{-1}$ for northeastern Japan and Nankai, respectively[36]. The $l$ value was calculated from the depth and horizontal distance of the thermal structures

in northeastern Japan and Nankai. We assumed that $h$ and $v$ are constant.

The thickness of sediment (including pore space) before subduction was 600–800 m and 1500–2250 m for the northeastern Japan and Nankai subduction zones, respectively[36,70]. The northernmost part of Japan is a non-accretionary subduction zone, whereas the Nankai region is an accretionary subduction zone, and the efficiency of sediment subduction differs between the two subduction zones[36]. We used the value of $h$ employed by Syracuse et al.[34] and van Keken et al.[28] ($h = 300$ m for the northeastern Japan and Nankai subduction zones), accounting for the effect of accretion at these active margins and the closure of pore space in the sediments during subduction[34]. Plank and Langmuir[35] estimated the thickness of subducting sediment to be 350 m for northeastern Japan ("Japan trench" in Plank and Langmuir[35]) and Nankai subduction zones. Using $h = 350$ m instead of 300 m resulted in little change in the calculated result.

### Calculating the mineral assemblage in the mantle wedge along the subduction megathrust

Based on mass balance considerations, the calculated $J$ can be used to estimate the amount of fluid infiltration into the mantle wedge ($\xi$; mol kg$^{-1}$ rock) as follows:

$$\xi = \frac{J}{\bar{m}\rho_{\mathrm{srp,0}}}\frac{t}{L}, \qquad (4)$$

where $\bar{m}$ (kg mol$^{-1}$) is the kg-molar-formula weight of the solvent, $\rho_{\mathrm{srp,0}}$ (kg m$^{-3}$) is the density of serpentinite at $\xi = 0$ (i.e., before fluid infiltration), $L$ (m) is the characteristic length, and $t$ (Myr) is the timescale. $\bar{m}$ and $\rho_{\mathrm{Srp,0}}$ were obtained using the WERAMI sub-program in Perple_X. Equation (4) assumes that all fluid released from the subducting metasediments is infiltrated into the forearc mantle.

To constrain the mineral assemblage in the mantle wedge at the calculated $\xi$, the modeled metasedimentary fluids were incrementally added to 1 kg of serpentinite using the 0-dimensional infiltration mode in Perple_X[39]. The fluid aliquot (0.1–1.0 mol) was added repeatedly in an isobaric–isothermal system to obtain a dataset containing the cumulative amount of fluid added and the modal abundance of minerals. By matching the calculated value of $\xi$ with the cumulative amount of fluid added, the mineral assemblage and its modal abundance at $\xi$ were obtained. No fluid or solid phases were fractioned (mode 0 in the VERTEX sub-program), but calculation with fluid fractionation (mode 1 in the VERTEX sub-program) was also conducted to investigate the effect on the model results (Supplementary Discussion 2). Calculations were made in the Si–Al–Fe–Mg–Ca–H$_2$–C–O$_2$ system. The whole-rock composition of the serpentinite was based on the global average of serpentinized harzburgite in a mantle wedge[71] (Supplementary Table 1). The whole-rock Fe$^{3+}$/∑Fe ratio was set to 0.58, similar to typical antigorite–serpentinite in subduction zones[64,65,72]. MATLAB scripts were developed to automatically generate input files and run the calculation of the 0-dimensional infiltration at each $P$–$T$ condition along the slab-top geotherm of the northeastern Japan and Nankai subduction zones.

The model results depend on the value of $t/L$. Therefore, when the timescale of fluid flow is independently constrained by geological evidence, the spatial scale of alteration at the base of the forearc mantle can be predicted. For example, for $t = 1$ Myr, the predicted mineral proportions at $t/L = 1$ Myr m$^{-1}$ (Fig. 2e–h) and 10 Myr m$^{-1}$ (Supplementary Discussion 2) represent those at the base of the mantle wedge in layers that are 1 m and 0.1 m thick, respectively. Similarly, for $t = 10$ Myr, the predicted mineral proportions at $t/L = 1$ (Fig. 2e–h) and 10 Myr m$^{-1}$ (Supplementary Discussion 2) can be regarded as the mineral assemblage at the base of the mantle wedge in layers that are 10 m and 1 m thick, respectively.

### Calculating the rate of carbon uptake in the Nankai subduction zone

Integrating the amount of carbonate minerals at 35–80 km depth in the Nankai subduction zone (Fig. 2h) yields a rate of cumulative carbon uptake in the mantle wedge of $5.4 \times 10^{-9}$ Mt C m$^{-1}$ yr$^{-1}$. Then, we assumed the same proportion of carbonate minerals in the entire subduction zone in southwestern Japan (900 km length)[36] to obtain a carbon uptake rate of 0.005 Mt C yr$^{-1}$.

## Data availability

The input and output files for the thermodynamic calculations are available from the data repository (https://doi.org/10.6084/m9.figshare.25771797)[73].

## Code availability

The software used for the thermodynamic calculations is freely available on the Perple_X website (https://www.perplex.ethz.ch/). The software MinVel is available from https://github.com/usgs/MinVel. MATLAB scripts used in this study will be made available on request.

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

## Acknowledgements

Ikuko Wada kindly provided the dataset for the subduction geotherm. We thank J.A.D. Connolly and the Perple_X community for continuous development and improvement of this software. Map figures were made in Cartopy Python package[74]. This research was supported by the Japan Society for the Promotion of Science KAKENHI grant numbers JP20KK0079 (RO), JP23K13194 (RO), JP22H05295 (AO), JP22H04932 (AO), and JP24H01011 (AO), and by Earthquake Research Institute (the University of Tokyo) Joint Research program ERI JURP 2024-B-01 (RO).

## Author contributions

R.O. and A.O. designed the study. R.O. conducted the thermodynamic calculations and processed the data. R.O. and A.O. contributed to the interpretation of the data. R.O. and A.O. prepared the draft manuscript. All authors were involved in revising the manuscript and approved the submitted version.

## Competing interests

The authors declare no competing interests.
