## [Peer Review File · Nature Communications]

REVIEWER COMMENTS

Reviewer #1 (Remarks to the Author):

Oyanagi & Okamoto employ the DEW molecular speciation model within the thermodynamic software Perple_X to investigate forearc mantle wedge carbonation and talc formation due to C-bearing sediment devolatilization during steady-state subduction. The authors focus on two slabtop PT profiles for the Nankai (warm) and NE Japan (cold) subduction margins. The findings suggest that talc + carbonate replacement of a hydrous ultramafic protolith (antigorite) occurs more readily along hot subduction margins and is less sensitive to the C mineralogy in the fluid-producing sediment (carbonate or reduced C).

The presence of talc within the forearc mantle wedge would have a significant control on the rheology within this region, and the reaction progresses more easily via introduction of C than by influx of Silica. This point has been made by previous authors, but the comparison between two nearby subduction margins is novel, as is the model approach.

The general premise and approach are fitting and the implications of the research are thought provoking. However, I find there to be moderate issues with the methodology, reporting of results, and the specific conclusions that are drawn. I feel that major revisions are required before this manuscript should be considered for publication.

I hope the authors find these comments constructive. Please find major comments and minor suggestions in the following text.

Major Comments:

The writing is difficult to follow at times, particularly in the introduction, and significant editing and rewriting is required.

The use of the term lithology should be changed throughout the manuscript. This is a somewhat ambiguous term. Consideration should be given to using a mineralogic or bulk compositional reference, some specific mentions are pointed out below.

Additionally, the structure of the manuscript could be redone. There are two relatively long sections: "Comparison with seismic observations" and "Consequences for the carbon cycle in the forearc mantle" that could be significantly altered. For the former, the seismic discussion could be reduced to a few sentences and reference to the extended figure, as the finding of this section is that carbonate + talc is indistinguishable from serpentinite and seismic investigation will not provide much help in identification of these zones. For the latter section (carbon cycling): this section actually mostly discusses rheologic implications, so should be renamed. Additionally, it seems a bit odd to focus a section on C cycling when the results suggest that it affects C subduction very little. An exception would be if the new results are at odds with previous studies (e.g., Stewart and Ague, 2020 nature comm).

The values for C content (L73-86) are extremely dependent on the model setup, and on which species

are excluded from the models. Without the full input and output files it is difficult to assess the validity of the reported values. This becomes particularly important when fractionating the fluid phase (as discussed in the Methods section), as the fractionation may lead to spurious results depending on if it was performed within Perple_X or as a post processing step (this latter point should be specified). A (supplementary?) figure detailing the C speciation and comparison with experiment/other models would be beneficial. Additionally (or alternatively) the Perple_X files (build, database, solution file, option file, P-T readin files) should be linked within the code/data availability statement. As it stands it is not possible to reproduce the work.

Likewise, the 0d infiltration model needs to be expanded on, and/or included in the data availability. Did you perform infiltration-fractionation or was fluid added to the antigorite and not removed?

If I understand correctly the sediment-derived fluid is interacted with a 1m thick hydrated mantle wedge peridotite until the fluid-rock ratio matches the value determined from the sediment devolatilization calculation. I see a few flaws in this methodology: (1) wouldn't the starting peridotite actually be anhydrous, at present you assume that serpentinization has already occurred; (2) how can the 1-m model be extrapolated to the whole of the mantle wedge; and (3) what is the rationale for "shutting off" the fluid-rock reaction. Points 2 and 3 are connected, if you extrapolate beyond a 1 meter thick layer you have to effectively continue to metasomatize the boundary layer (i.e., the initial 1 meter thick layer). In other models (and experiment) this fluid dominated peridotite metasomatism leads to a talc-free assemblage and production of qtz + carbonate (e.g., Sieber et al., 2022 CMP 177, 44).

The use of GLOSS in the supplementary sensitivity analysis should be reevaluated. GLOSS represents an average of all subducted sediment but is a poor choice of subducted composition for almost any margin. Changing the C-content of GLOSS without also adjusting other elements (Ca, K, Na, Mg, Fe) will lead to results that do not have much meaning. A better approach would be to perform this analysis on a set of compositions that correspond to the typical subducted sediment endmembers. Following the method of Hacker (2008) and van Keken et al., (2011) this could be terrigenous, turbidite, pelagic, diatomaceous ooze, chert, and carbonate. I do not think that the GLOSS assessment adds too much to the present work.

The assumption of 300m of sediment thickness may be a major control on the results in the present work. What is the rationale for taking two sediment section which differ in thickness by a factor of ~3 (800 and 2300m) and reducing them to the same thickness of 300m? I think this should be rationalized.

Minor Comments:

L35: "involving" is grammatically incorrect in this sentence.

L37-40: check grammar: Subducted carbon "is" mobilized.

L46-48: this may be the case as it pertains to carbon, but in general the progressive mineralogic changes have been discussed (e.g. Peacock, 1996, Subduction Top to Bottom; Abers et al., 2017 Nature Geosci.;

Till et al., 2012 CMP)

L48-52: confusing and wordy sentence. Consider removing geological constraints, and separating into two sentences, the last part of the sentence does not fit.

Introduction: This section contains a number of grammatical errors that make it difficult to read. Many sentences contain multiple, disconnected statements (e.g., L43-44, L48-52). At present this section does not serve to properly introduce the importance of the study.

L 56 and elsewhere: Consider changing lithology throughout, it is not a very specific term (e.g., the mantle wedge is almost always an ultramafic lithology). Consider instead “variations in mantle wedge bulk composition” and mineralogic changes associated with open system fluid rock interaction.

L55-57: A bit more information would be useful here and would not add to length extensively: e.g., “we utilize gibbs free energy minimization of mineral solid solutions and electrolyte speciation within a pseudo-open system to determine chemical mass transfer during fluid-rock interaction.”

L58-59: You are predicting the progressive change of some initial bulk composition here, not predicting the lithology. Again, I suggest you use bulk composition in place of lithology.

L61: enhances the precision? I am not sure what this means in the present usage? Please expand.

L98-101: some discussion should be devoted to how the results for a 1 meter thick layer are extrapolated to the whole of the mantle wedge. As fluid percolates further into the mantle wedge the 1 meter region will undergo more extensive fluid-rock interaction potentially to the point of stabilizing a quartz-carbonate assemblage, and removing talc from the system entirely.

L115: reveals an evolving bulk composition. unclear.

Equations 1 and 2: Specifically this is the chemical formula of antigorite, this should be noted. For NE Japan, the PT conditions may actually stabilize lizardite, some discussion is warranted.

Equation 2: Formula is simpler if written as 2 serpentine + 45 CO₂  17 Talc . . . etc..

L146: GLOSS with modified TC contents is an inappropriate methodology. Speciation is going to be dependent on other chemical elements present (e.g., Na, K, Ca, Fe, Mg, Mn). GLOSS is an average of all subducted sediment, and does not do a particularly good job of matching any margins sediment input. My suggestion would be to redo and expand this calculation instead using the 5-6 most common subducted sedimentary compositions: e.g., terrigenous, turbidite, pelagic, diatomaceous ooze, chert, and carbonate (after van Keken et al., 2011 for example).

L168-187: This section could be greatly reduced to a sentence or two, as the main conclusion here is that we cannot seismically distinguish serpentine from talc-carbonate.

L189-247: this section is mostly about rheologic controls of talc as opposed to carbon cycling. As mentioned, forearc C release does not greatly alter the C cycling. Consider renaming this section to focus on the (more important) role of talc in controlling rheology. Again, the C cycling discussion could probably be reduced to a few sentences.

Fig. 1a: need scale.

Fig. 1d: consider PT paths for all of the subduction zones discussed and labelled in panels b and c.

L473: Organic species: a somewhat ambiguous statement. Referring to Methane, carbonic acid, and other C-H species? Or just FORMATE, ACETATE, etc. . .

L516: was excess fluid fractionated? or only added to the serpentinite.

L540-541: this should also include the build files, thermodynamic database, solid solution file, and perplex_option_files for the present work.

I hope these comments are useful!
Sincerely,

Reviewer #2 (Remarks to the Author):

Please see my comments in the attachment. I hope my suggestions are helpful, and that my comments allow you to improve the manuscript.

Reviewer #3 (Remarks to the Author):

The authors investigate the influence of mantle wedge fluid transport on spatially variable subduction zones. For this purpose, they use Perple_X, which is a commonly used thermodynamic modeling software. The authors specifically look at the subduction zones in Northeast and Southeast Japan, which have a different thermal structure and chemical compositions according to the authors. The authors conclude that the infiltration of carbon-bearing fluids causes heterogeneities in the mantle wedge, which could have an effect on the mechanical properties of the subduction zone.

The paper is interesting, generally thorough, and well explained, with an important exception for the methods as stated below, which requires clarification. I am not able to fully review the geochemistry of the paper, since that is outside my field of expertise.

In the methods sections 2 and 3 (lines 492 to 529), two equations are stated to be used. The authors

explain what the individual elements of the equation do and how it is used, but it is not explained where these equations come from and what their assumptions are. This needs to be addressed.

In the code availability section, MinVel is not mentioned, while it is used. It is not clear to me whether the equations in the method section are computed by hand, or are part of a workflow of scripts and/or code. If code is used, it should be mentioned here. It would also be useful to be more clear about the actual workflow in the methods section.

Looking at the submission requirements (<https://www.nature.com/ncomms/submit/article>) state: "The Results and Methods sections should be divided by topical subheadings; the Discussion should be succinct and may not contain subheadings." The results do not contain subheadings, which is understandable in this case because it is short, although a clearer distinction between the setup description and the actual results could be useful. An alternative is to move the first paragraph of the results up to the introduction. The Methods contain subheadings and this seems appropriate. The Discussion also contains subheadings, which are appropriate for the length but don't conform with the submission requirements. I will leave this up to the editor.

Author replies to reviewer's comments

(Author replies in blue)

Reviewer #1

Oyanagi & Okamoto employ the DEW molecular speciation model within the thermodynamic software Perple_X to investigate forearc mantle wedge carbonation and talc formation due to C-bearing sediment devolatilization during steady-state subduction. The authors focus on two slab top PT profiles for the Nankai (warm) and NE Japan (cold) subduction margins. The findings suggest that talc + carbonate replacement of a hydrous ultramafic protolith (antigorite) occurs more readily along hot subduction margins and is less sensitive to the C mineralogy in the fluid-producing sediment (carbonate or reduced C). The presence of talc within the forearc mantle wedge would have a significant control on the rheology within this region, and the reaction progresses more easily via introduction of C than by influx of Silica. This point has been made by previous authors, but the comparison between two nearby subduction margins is novel, as is the model approach. The general premise and approach are fitting and the implications of the research are thought provoking. However, I find there to be moderate issues with the methodology, reporting of results, and the specific conclusions that are drawn. I feel that major revisions are required before this manuscript should be considered for publication. I hope the authors find these comments constructive. Please find major comments and minor suggestions in the following text.

We appreciate the reviewer's comments, which indicate that our methods and implications are effective and thought-provoking.

Major Comments:

The writing is difficult to follow at times, particularly in the introduction, and significant editing and rewriting is required.

In the revised version of the manuscript, we carefully improved the clarity and coherence of the manuscript by addressing grammatical issues and ensuring that the sentences flow logically.

The use of the term lithology should be changed throughout the manuscript. This is a somewhat ambiguous term. Consideration should be given to using a mineralogic or bulk compositional reference, some specific mentions are pointed out below.

We agree with this. After careful consideration, we used "mineral assemblage" instead the term. We corrected this point throughout the manuscript.

Additionally, the structure of the manuscript could be redone. There are two relatively long sections: "Comparison with seismic observations" and "Consequences for the carbon cycle in the forearc mantle" that could be significantly altered. For the former, the seismic discussion could be reduced to a few sentences and reference to the extended figure, as the finding of this section is that carbonate + talc is indistinguishable from serpentinite and seismic investigation will not provide much help in identification of these zones.

As suggested by the reviewer #1, we reduced sentences in the section "Comparison with seismic observations".

For the latter section (carbon cycling): this section actually mostly discusses rheologic implications, so should be renamed. Additionally, it seems a bit odd to focus a section on C cycling when the results suggest that it affects C subduction very little. An exception

would be if the new results are at odds with previous studies (e.g., Stewart and Ague, 2020 nature comm).

We agree with this suggestion.

The section was renamed as suggested. The section title is “Rheological consequences of heterogeneous talc formation along a subduction megathrust”.

In addition, as suggested, we shortened the discussion of forearc C uptake and the separated section has four sentences (line 274). To reduce the number of sentences, we move some sentences to the METHOD section ("Calculating the rate of carbon uptake in the Nankai subduction zone") (line 435).

The values for C content (L73-86) are extremely dependent on the model setup, and on which species are excluded from the models. Without the full input and output files it is difficult to assess the validity of the reported values.

In the revised manuscript, we provided input and output files. See the “DATA AVAILABILITY” section. We note that related files for NE Japan and Nankai sediments are at folder “1_Sediments in NE Japan” and “2_Sediments at Nankai”, respectively.

This becomes particularly important when fractionating the fluid phase (as discussed in the Methods section), as the fractionation may lead to spurious results depending on if it was performed within Perple_X or as a post processing step (this latter point should be specified).

Our calculation did not fractionate the solid and fluid phases (fluid was only added to the serpentinite). This is clarified in line 415.

However, we compared the model prediction between the batch and fractional calculations (Supplementary Discussion 2). In this case, the fractionation was performed by the VERTEX software (mode 1 calculation). The comparison showed that the predicted results were not significantly changed.

A (supplementary?) figure detailing the C speciation and comparison with experiment/other models would be beneficial. Additionally (or alternatively) the Perple_X files (build, database, solution file, option file, P-T readin files) should be linked within the code/data availability statement. As it stands it is not possible to reproduce the work. Likewise, the 0d infiltration model needs to be expanded on, and/or included in the data availability.

We followed this comment, and the thermodynamic database, solid solution file, and perplex_option_file are available in the repository. We mentioned this point in the DATA AVAILABILITY section.

Did you perform infiltration-fractionation or was fluid added to the antigorite and not removed?

Our calculation did not fractionate the solid and fluid phases (fluid was only added to the serpentinite). This is clarified in line 415.

However, we compared the model prediction between the batch and fractional calculations (Supplementary Discussion 2). In this case, the fractionation was performed by the VERTEX software (mode 1 calculation). The comparison showed that the predicted results were not significantly changed.

If I understand correctly the sediment-derived fluid is interacted with a 1m thick hydrated mantle wedge peridotite until the fluid-rock ratio matches the value determined from the sediment devolatilization calculation. I see a few flaws in this methodology: (1) wouldn't the starting peridotite actually be anhydrous, at present you assume that serpentinization has already occurred;

We conducted new calculations, examining the variation of mineral assemblage in the anhydrous forearc mantle (i.e., peridotite) in the response to the fluid infiltration from the subducting sediments. The predicted results using anhydrous mantle were similar with the predicted results using hydrated one (i.e., serpentinite). In the revised manuscript, we added sentence describing the model result at line 176. Main results of the calculation using anhydrous mantle were described in the Supplementary discussion 3.

(2) how can the 1-m model be extrapolated to the whole of the mantle wedge; and (3) what is the rationale for “shutting off” the fluid-rock reaction. Points 2 and 3 are connected, if you extrapolate beyond a 1 meter thick layer you have to effectively continue to metasomatize the boundary layer (i.e., the initial 1 meter thick layer). In other models (and experiment) this fluid dominated peridotite metasomatism leads to a talc-free assemblage and production of qtz + carbonate (e.g., Sieber et al., 2022 CMP 177, 44).

First, our model did not “shutting off” the fluid-rock reaction. The expression used in the previous manuscript may be misleading.

The prediction of mineral assemblage in our model depends on the parameter t/L , which is the ratio of the time scale (Myr) relative to the scale length (m) of fluid infiltration (thickness of metasomatized layer). When either t or L is constrained or assumed, the other is determined. Therefore, our model results are applicable to any t and L of interest. (line 425).

The results calculated with t/L can be interpreted clearly when the timescale of fluid-rock interaction is independently constrained by the geologic evidences. For example, the tectonic constraints of the Nankai subduction zone suggest that the subduction of the Philippine Sea Plate (re)initiated at 17 Ma. Therefore, we assumed that timescale of fluid infiltration of 1–10 Myrs. Therefore, our model results represent

- After 1 Myr of fluid infiltration, our prediction suggests that a 1-m-thick layer at the base of the forearc mantle was composed of talc-poor serpentinite around the MWC and talc + carbonate rocks at greater depth (based on the results from $t/L = 1 \text{ Myr m}^{-1}$; Figs. 2g, h and 3a-c).
- After 10 Myr of fluid infiltration, our prediction suggests that the 1 m thick basement layer was transformed to talc + carbonate at 35-50 km to quartz + magnesite assemblage at 50-74 km (based on the results from $t/L = 10 \text{ Myr m}^{-1}$; Supplementary Discussion 2; Fig. 3d, e), but a 10 m thick basement layer composed of serpentinite around MWC and talc + carbonate rocks at greater depth was expected to occur simultaneously when continuous fluid infiltration occurs beyond the 1 m thick layer (based on the results from $t/L = 1 \text{ Myr m}^{-1}$; Figs. 2g, h and 3d, e).

We have added these discussions (on lines 141–144 and 164 –173) and Figure 3b-e in the revised manuscript.

Although the talc-poor assemblage (carbonate + quartz) was predicted at the greater depth, we emphasize that the rheological contrast between MWC and the greater depth is still valid because a thicker layer rich in talc + carbonate rock was also predicted. This point is discussed in line 319.

The use of GLOSS in the supplementary sensitivity analysis should be reevaluated. GLOSS represents an average of all subducted sediment but is a poor choice of subducted composition for almost any margin. Changing the C-content of GLOSS without also adjusting other elements (Ca, K, Na, Mg, Fe) will lead to results that do not have much meaning. A better approach would be to perform this analysis on a set of compositions that correspond to the typical subducted sediment endmembers. Following the method of Hacker (2008) and van Keken et al., (2011) this could be terrigenous, turbidite, pelagic, diatomaceous ooze, chert, and carbonate. I do not think that the GLOSS assessment adds too much to the present work.

We appreciate these suggestions, which were helpful in strengthening our claim.

First, we suggest that the calculation (GLOSS with modified TC content) has implications on the role of sediment input on talc formation, as it showed that (i) TC and F_{OC} are less effective at the warm subduction zone, but effective at the cold subduction zone (Supplementary Discussion 7).

Nevertheless, in the revised manuscript, we followed the suggestion and further sensitivity analyses were performed using the bulk rock composition of the sediments. The bulk rock composition of sediments used in van Keken et al. (2011) was not used here because it does not contain C. Alternatively, we used the bulk rock composition of carbonate sediments, chert, pelagic clay, terrigenous sediments, and turbidite from Plank and Langmuir (1998). The results of the sensitivity analysis were summarized as follows;

- The model prediction shows carbonate sediment and chert may release minimal H₂O to the forearc mantle wedge, as previously suggested by Hacker (2008) and van Keken et al. (2011). Therefore, dehydration from subducting carbonate sediment and chert may not result in the talc formation in the forearc mantle.
- For subduction of pelagic clay, terrigenous sediments, and turbidite, the amount of fluid required for 1 vol% talc is generally low in the Nankai subduction zone compared to those in NE Japan. Therefore, talc would be formed predominantly in the warm subduction zone, rather than the cold subduction zone.
- For subduction of pelagic clay, terrigenous sediments and turbidite in both NE Japan and Nankai subduction zones, ξ_{Tlc} increases in the order of pelagic clay, turbidite, and terrigenous sediment.

In the main manuscript, we added the sentences to briefly explain the result of the

sensitivity analysis on line 220 – 225. In addition, the complete results of the parameter studies were described in Supplementary Discussion 7.

The result of the parameter studies does not affect the conclusion of the manuscript based on the calculations for the NE Japan and Nankai subduction zones. However, the inferred talc occurrence in other subduction zones can be changed. For example, the subducting sediments in the Colombia subduction zone are the carbonate sediments (Plank 2014), but the parameter study suggests that subduction of carbonate sediments causes less talc formation. We added this discussion on line 254.

References

- Hacker, B. R. H₂O subduction beyond arcs. *Geochemistry, Geophysics, Geosystems*, 9 (2008).
- Plank, T. & Langmuir, C. H. The chemical composition of subducting sediment and its consequences for the crust and mantle. *Chem. Geol.* 145, 325–394 (1998).
- Plank, T. *The Chemical Composition of Subducting Sediments. in Treatise on Geochemistry* 4, 607–629 (Elsevier, 2014).
- Syracuse, E. M. et al. The global range of subduction zone thermal models. *Phys. Earth Planet. Inter.* 183, 73–90 (2010).
- van Keken, P. E., Hacker, B. R., Syracuse, E. M. & Abers, G. A. Subduction factory: 4. Depth-dependent flux of H₂O from subducting slabs worldwide. *J. Geophys. Res.* 116, B01401 (2011).

The assumption of 300m of sediment thickness may be a major control on the results in the present work. What is the rationale for taking two sediment section which differ in thickness by a factor of ~3 (800 and 2300m) and reducing them to the same thickness of 300m? I think this should be rationalized.

The Nankai subduction zone is an accretion subduction zone, whereas the northeast Japan is an erosive subduction zone. In addition, the sediment is porous before subduction. Therefore, accretion and compaction change the thickness of the subducted sediments, and the proportion of subducting and accreting sediments would vary in the subduction zone.

In our calculations, we used the values of sediment thickness used in Syracuse et al. (2010) and van Keken et al. (2018). The values of sediment thickness used in these studies account for the effects of compaction and accretion of subducting sediments. As another estimate, Plank and Langmuir (1998) estimated these to be 350 m for the northeastern Japan and Nankai subduction zones, respectively. Using the value (350 m) results in ~1.2 times higher fluid flux, but the model prediction would not produce large differences.

In the revised manuscript, we mentioned these points in line 387–397.

References

- Plank, T. & Langmuir, C. H. The chemical composition of subducting sediment and its consequences for the crust and mantle. *Chem. Geol.* 145, 325–394

(1998).

Syracuse, E. M. et al. The global range of subduction zone thermal models. *Phys. Earth Planet. Inter.* 183, 73–90 (2010).

van Keken, P. E., Wada, I., Abers, G. A., Hacker, B. R., & Wang, K. Mafic high-pressure rocks are preferentially exhumed from warm subduction settings. *Geochemistry, Geophysics, Geosystems*, 19(9), 2934-2961 (2018).

Minor Comments:

L35: “involving” is grammatically incorrect in this sentence.

Corrected. Thank you.

L37-40: check grammar: Subducted carbon "is" mobilized.

Thank you for pointing this out. Corrected. Moreover, the revised manuscript has been thoroughly reviewed by an editing company to minimize any inconsistencies or grammatical errors.

L46-48: this may be the case as it pertains to carbon, but in general the progressive mineralogic changes have been discussed (e.g. Peacock, 1996, Subduction Top to Bottom; Abers et al., 2017 Nature Geosci.; Till et al., 2012 CMP)

We thank this. In the revised manuscript, we reviewed previous modeling of the progressive mineralogic changes, as follows (Line 51);

“Based on petrological thermodynamic models, the interaction of slab-derived fluid with overlying mantle peridotite has been studied for pure H₂O^{21,23,24} or H₂O-rich fluids that contain dissolved rock components, but without considering aqueous carbon^{25–28}. In contrast, the interaction of rock with fluid containing aqueous carbon and dissolved rock components has been investigated for specific *P–T* conditions¹⁰ or several sets of *P–T* conditions along a subduction geotherm²⁹.”

L48-52: confusing and wordy sentence. Consider removing geological constraints, and separating into two sentences, the last part of the sentence does not fit.

As suggested, we modified the sentence. Moreover, in the revised manuscript, the sentence was thoroughly rewritten.

Introduction: This section contains a number of grammatical errors that make it difficult to read. Many sentences contain multiple, disconnected statements (e.g., L43-44, L48-52). At present this section does not serve to properly introduce the importance of the study.

In the revised version of the manuscript, the INTRODUCTION section has been thoroughly rewritten to emphasize the significance of the present study. Additionally, the manuscript has been thoroughly reviewed by an editing company to minimize any inconsistencies or grammatical errors.

L 56 and elsewhere: Consider changing lithology throughout, it is not a very specific term (e.g., the mantle wedge is almost always an ultramafic lithology). Consider instead “variations in mantle wedge bulk composition” and mineralogic changes associated with open system fluid rock interaction.

We agree. After careful consideration, we used the term "mineral assemblage" instead. We corrected this point throughout the manuscript.

L55-57: A bit more information would be useful here and would not add to length extensively: e.g., “we utilize gibbs free energy minimization of mineral solid solutions and electrolyte speciation within a pseudo-open system to determine chemical mass transfer during fluid-rock interaction.”

We modified this as follows (line 65);

“Minimization of the Gibbs free energy of mineral solid solutions and electrolyte speciation within a pseudo-open system³⁵ was used to constrain chemical mass transfer during fluid–rock interaction.”

L58-59: You are predicting the progressive change of some initial bulk composition here, not predicting the lithology. Again, I suggest you use bulk composition in place of lithology.

We agree. After careful consideration, we used the term “mineral assemblage” instead. We corrected this point throughout the manuscript.

L61: enhances the precision? I am not sure what this means in the present usage? Please expand.

We clarified this point, as follows (line 67);

“In the modelling, all of the major rock components (e.g., C, SiO₂, and Al₂O₃) were considered to be soluble in the fluid, to be consistent with geologically observed mass transfer across the slab–mantle interface^{12,13,36}.”

L98-101: some discussion should be devoted to how the results for a 1 meter thick layer are extrapolated to the whole of the mantle wedge.

In the revised manuscript, we emphasized that the model considers the assemblage of the base of mantle wedge.

As fluid percolates further into the mantle wedge the 1 meter region will undergo more extensive fluid-rock interaction potentially to the point of stabilizing a quartz-carbonate assemblage, and removing talc from the system entirely.

As reviewer suggested, quartz + carbonate rich rock is formed at the greater depth in the Nankai subduction zone, but talc + carbonate rich rocks are also formed at the same depth with over the 1 meter region. The thickness of quartz + carbonate layer is much less than those of talc + carbonate layer at the same depth. Therefore, our rheological implications were not influenced (Fig. 3). See above reply for details.

L115: reveals an evolving bulk composition. unclear.

Thank you for pointing this out. We used “mineral assemblage” instead “lithology”.

Equations 1 and 2: Specifically this is the chemical formula of antigorite, this should be noted.

We noted that serpentine is antigorite in these equations (lines 187 and 191).

For NE Japan, the PT conditions may actually stabilize lizardite, some discussion is warranted.

Following sentences were added (line 192).

“In reactions (1) and (2), the serpentine composition is represented by antigorite.

Although the cold geotherm of the northeastern Japan subduction zone might result in the stability of lizardite or chrysotile rather than antigorite, the reactions would be similar to those shown.”

Equation 2: Formula is simpler if written as 2 serpentine + 45 CO₂  17 Talc . . . etc..

We modified so.

L146: GLOSS with modified TC contents is an inappropriate methodology. Speciation is going to be dependent on other chemical elements present (e.g., Na, K, Ca, Fe, Mg, Mn). GLOSS is an average of all subducted sediment, and does not do a particularly good job of matching any margins sediment input. My suggestion would be to redo and expand this calculation instead using the 5-6 most common subducted sedimentary compositions: e.g., terrigenous, turbidite, pelagic, diatomaceous ooze, chert, and carbonate (after van Keken et al., 2011 for example).

First, we suggest that the calculation (GLOSS with modified TC content) has implications on the role of sediment input on talc formation, as it showed that (i) TC and F_{OC} are less effective at the warm subduction zone, but effective at the cold subduction zone (Supplementary Discussion 7).

Nevertheless, we have followed this comment and performed further calculations using five representative compositions of subducting sediments: carbonate sediments, chert, pelagic clay, terrigenous sediments, and turbidite.

The results of the parameter studies do not affect the conclusions of the manuscript. However, the inferred talc occurrence in other subduction zones has been modified. The results of the parameter study suggest that subduction of carbonate sediments results in less talc formation. The subducting sediments in the Colombia subduction zone are the carbonate sediments (Plank 2014), so less talc would occur in the Colombia subduction zone.

See response above for details.

References

Syracuse, E. M. et al. The global range of subduction zone thermal models. *Phys. Earth Planet. Inter.* 183, 73–90 (2010).

L168-187: This section could be greatly reduced to a sentence or two, as the main conclusion here is that we cannot seismically distinguish serpentine from talc-carbonate.

In the revised manuscript, we made an effort to minimize the number of sentences as much as feasible. However, we concluded that the sentence regarding seismic observations from previous research could not be shortened further. In contrast we shortened the sentence describing our calculation of seismic properties. Consequently, the paragraph contains five sentences in the revised manuscript (lines 260 – 270).

L189-247: this section is mostly about rheologic controls of talc as opposed to carbon cycling. As mentioned, forearc C release does not greatly alter the C cycling. Consider renaming this section to focus on the (more important) role of talc in controlling rheology.

Again, the C cycling discussion could probably be reduced to a few sentences.

The section was renamed to “Rheological consequences of heterogeneous talc formation along a subduction megathrust”.

Moreover, as suggested, we have shortened the discussion of forearc C uptake and the separated section has four sentences (line 274). To reduce sentence, we move some sentence into METHOD section (“Calculating the rate of carbon uptake in the Nankai subduction zone.”) (line 435).

Fig. 1a: need scale.

Human in the Fig. 1a is scale. We added scale bar to increase readability (Fig. 1a).

Fig. 1d: consider PT paths for all of the subduction zones discussed and labelled in panels b and c.

We added PT paths for all subduction zones labeled in b (Fig. 1c). These are all from Wada and Wang (2009). We renamed the Izu-Bonin and Tonga subduction zones to Izu and Kermadec subduction zones to be consistent with Wada and Wang (2009).

Reference

Wada, I. & Wang, K. Common depth of slab-mantle decoupling: Reconciling diversity and uniformity of subduction zones. *Geochemistry, Geophys. Geosystems* 10, (2009).

L473: Organic species: a somewhat ambiguous statement. Referring to Methane, carbonic acid, and other C-H species? Or just FORMATE, ACETATE, etc. . .

Just FORMATE, ACETATE, etc... Moreover, in the revised manuscript, the model setup would be more recognized in detail, because we provided full input and output files in the Data Availability section.

In the revised manuscript, we clarified these points, as follows;

Line 355: “High molecular weight organic compounds (e.g., HCOO⁻ and CH₃COO⁻) were excluded from the computations”

Line 358: “(for details, see the input files in the Data Availability section).”

L516: was excess fluid fractionated? or only added to the serpentinite.

In our calculation, no solid and fluid phases were fractionated (fluid were only added to the serpentinite). This is clarified at line 415.

L540-541: this should also include the build files, thermodynamic database, solid solution file, and perplex_option_files for the present work.

As suggested, the build files, thermodynamic database, solid solution file, and perplex_option_files were available from the repository. We mentioned this point in the “Data availability” section.

I hope these comments are useful! Sincerely, Gabe S. Epstein

We appreciate Dr Epstein for constructive reviews and suggestions, which were greatly helpful to improve our manuscript.

Reviewer #2

This manuscript presents the results of state-of-the-art thermodynamic modelling of contrasting cold and warm subduction zones in Japan to investigate the role of metasediment dehydration and carbon solubility in carbon-bearing subducted metasediments on the formation of talc in the overlying mantle wedge at the plate interface. This is important because talc has been suggested by previous studies to exert control on the rheological properties of the subduction plate interface. The thermodynamic modelling presented uses as inputs the site-specific bulk compositions of sediments on each of the downgoing plates of the respective subduction zones, taking into account their total carbon content, as well as the amount of organic carbon vs. carbonate-hosted (inorganic) carbon. The study presented here suggests: 1) talc can be produced at the plate interface through aqueous, CO₂-bearing fluid infiltration into ultramafic material, with the fluid derived from the dehydration of carbon-bearing metasediments (and attendant carbon dissolution in the fluid). 2) rheologically significant amounts of talc are predicted to be produced at the plate interface in a warm subduction zone (Nankai), at depths that correspond to the aseismic portion of the plate interface. 3) Talc production at the plate interface in a cold subduction zone (NE Japan) is limited to the lower end of the hypothesized Maximum Decoupling Depth. 4) Points 2 and 3 above suggest that infiltration of metasedimentary-derived, C-bearing aqueous fluids may thus exert strong control on the rheological and seismic behavior of the plate interface. The figures are polished and easy to read. These conclusions are interesting and potentially significant to a broader audience, and are thus suitable for publication in *Nature Communications*. I raise some concerns and suggestions here for the authors to improve the manuscript.

We are grateful for the reviewer's high evaluation of our manuscript and for the comments and suggestions that helped to improve the manuscript.

Major comments/concerns/suggestions:

Why did you use the thermal models from Wada and Wang (2009) over others? If there is no justification for choosing only one set of thermal models, it would be good to show that the results are the same, regardless of which thermal models are chosen.

This is because the Wada and Wang (2009) thermal model provides a site-specific thermal profile around the seismogenic zone. We consider that other available thermal models (Syracuse et al. 2010; van Keken et al. 2011) would not be suitable for the purpose of the present study because they are not site-specific but averaged over areas of hundreds of kilometers.

In the revised manuscript, we mentioned this point, as follows:

“We used the slab-top geotherms calculated by Wada and Wang²² because they are site-specific geotherms that cross the boundary between the seismogenic and aseismic zones, rather than using the average geotherm over an area of hundreds of kilometers in scale.” (Line 81).

References

- Syracuse, E. M. et al. The global range of subduction zone thermal models. *Phys. Earth Planet. Inter.* 183, 73–90 (2010).
- van Keken, P. E., Hacker, B. R., Syracuse, E. M. & Abers, G. A. Subduction factory: 4. Depth-dependent flux of H₂O from subducting slabs worldwide. *J. Geophys. Res.* 116, B01401 (2011).

Wada, I. & Wang, K. Common depth of slab-mantle decoupling: Reconciling diversity and uniformity of subduction zones. *Geochemistry, Geophys. Geosystems* 10, (2009).

The introduction still needs a sentence that gives the reader a broad overview of what you did and why you did it....perhaps the first sentence of the results section, or a variation of the sentence in lines 20-24 of the abstract (“Here, we investigated...”) can be moved to the introduction.

We thank this suggestion, which contributed enhancing readability of this manuscript.

In the revised manuscript, we added sentences describing overview. Line 64–76.

Why do you only take into account the fluid released by dehydration of metasediments? Subducting meta-mafic rocks (basalt) have been shown to have a significant contribution to the overall water budget at the plate interface over the depth range you are considering, and to be correlated to the occurrence of slow slip earthquakes (e.g. Condit et al., 2020). We also know that altered MORB can contribute significant CO₂ (e.g., Gorce et al., 2019). So, this brings the question: how would the flux of aqueous and CO₂-bearing fluids derived from underlying metabasalts that then infiltrate into the overlying sediments affect the dissolved C in the fluid, and thus the potential to produce talc in ultramafic lithologies at the plate interface?

We appreciate this comment, which helped to improve and strengthen our paper.

As suggested by this comment, the hydrous oceanic crust dehydrates during subduction, and the generated fluids may be infiltrated into the overlying metasediments. Unfortunately, it is currently difficult to model the coupled process of fluid-rock equilibria and electrolyte fluid transport for three lithologies; the subducting metasediments, the oceanic crust, and mantle wedge. Here, following Lindquist et al. (2023), we have added simple calculations that consider both dehydration from metasediments and altered oceanic crust (AOC). The fluid flux from subducting AOC was calculated, then the fluid flux from subducting oceanic crust and metasediments was summed. The summed fluid flux was then used to estimate the mineralogy of the forearc mantle.

The calculation with AOC suggests that AOC dehydrates at (i) > 73 km depth in northeastern Japan and (ii) 29-32 km and >58 km depth in the Nankai subduction zone, suggesting that AOC contributes additional fluid input at depth. Using the summed fluid flux, the predicted mineralogy shows a greater amount of talc + carbonate at >58 km depth in the Nankai subduction zone. Interestingly, even when considering fluids from both AOC and metasediments, the downdip variation of talc is not significantly changed because AOC dehydration may not occur at the mantle wedge corner at Nankai (35 km depth).

In summary, our additional calculations suggest that fluids from the AOC contribute to increased fluid flux at >58 km depth in the Nankai subduction zone, resulting in more pronounced heterogeneities in talc occurrence between the mantle wedge corner and deeper forearc mantle.

We discussed these points at line 178–182 in the main manuscript. The methodology and full description of the model prediction of dehydration from the AOC are provided in Supplementary Discussion 4.

It might be good to include a discussion on the effects of underplating – it seems this is a process that is commonly invoked for the removal of slab material (and to keep slab material such as sediments from subducting further). If sediment underplating is a common process, then perhaps that places a limit on the depth to which your preferred mechanism of talc production is viable.

Thank you for this insightful comment. After careful consideration, we suggest that sediment underplating may not affect our preferred mechanism of talc production.

On-land exposures of the paleo-subduction zone suggest that the proportion of underplated sediments is variable, ranging from 20–85% of subducted sediments (e.g., Clift and Hartley 2007; Tewksbury-Christle et al. 2021). As the proportion of underplated sediments increases, fluid flux from subducting sediments to the overlying forearc mantle would be reduced, and AOC dehydration would become the dominant source of fluid to the forearc mantle.

In the Nankai subduction zone, the calculation with AOC showed that the fluid fluxes around the MWC (35 km) and at greater depths (>58 km) were dominated by those from subducting sediments and AOC, respectively (Supplementary Fig. 5d). Therefore, as the proportion of subducting sediments increases, fluid fluxes around the MWC may decrease, while those at greater depth may show less change. These depth variations in fluid flux associated with sediment underplating would cause more contrasting heterogeneities in talc distribution.

We added these discussions on the effect of underplating (Supplementary Discussion 5).

References

- Clift, P. D. & Hartley, A. J. Slow rates of subduction erosion and coastal underplating along the Andean margin of Chile and Peru. *Geology* 35, 503–506 (2007)
- Tewksbury-Christle, C. M., Behr, W. M. & Helper, M. A. Tracking Deep Sediment Underplating in a Fossil Subduction Margin: Implications for Interface Rheology and Mass and Volatile Recycling. *Geochemistry, Geophys. Geosystems* 22, 1–23 (2021)

Why did you restrict your modeling to the two Japan subduction zones? Why not compare and contrast several warm and cold subduction zones using the same methodology? Perhaps there was good reason for this, but it was not clear to me in the manuscript.

We added following sentences (line 79–81).

“Thermodynamic calculations were conducted for two contrasting subduction megathrusts in northeastern Japan and Nankai (southwestern Japan; Fig. 1b, c). These regions are considered to be representative cold and warm subduction zones, respectively^{33,37}.”

In the revised manuscript, we also applied the same methodology to another

subduction zone, Cascadia. The model results were similar to those of the Nankai subduction zones.

In the revised manuscript, we added a section "Changes in mineral assemblage in other subduction zones" to discuss the model results in addition to NE Japan and Nankai (line 231–237). We also added the description of the model result for the Cascadia subduction zone (Supplementary Discussion 9).

There are several instances in which it is important to distinguish the results predicted by your modelling. Otherwise, it could be interpreted by a reader as fact (without a citation). Some examples from the text include: Line 73-74: "the fluid equilibrated with metasedimentary rocks is rich in Na...and Si..." This should be changed to "the fluid predicted by our modelling to be in equilibrium with..." Line 93-94: "dehydration from subducting sediments occurs at depths of 20-30 km..." This can be interpreted by a reader as a truth that is geophysically observed....but it is only predicted by your modelling! Line 103: "minerals and carbonates are not formed"....again, this is something that is predicted by your modelling, not something that has been observed. Line 119-120 "The occurrence of talc without carbonate was locally observed at a depth of 70 km" No, it was not! There are many more examples of this – please read through your manuscript carefully and modify as appropriate.

Thank you for pointing this out. We agree. We read the manuscript carefully and wrote the model predictions and geophysical observations differently.

Line 100: It is unclear why you chose a 1 Myr duration of fluid infiltration into the mantle wedge. What was the justification for this? If it's arbitrarily chosen, you need to show that other timescales would have similar model results.

The expression used in the previous manuscript may be misleading.

The prediction of mineral assemblage in our model depends on the parameter t/L , which is the ratio of the time scale (Myr) relative to the scale length (m) of fluid infiltration. When either t or L is constrained or assumed, the other is determined. Therefore, our model results are applicable to any t and L of interest. (line 425).

Previous manuscript showed the predicted results at $t/L = 1$. In the revised manuscript, we discussed variation of model prediction at $t/L = 0.1, 1, \text{ and } 10$. The results from the batch model calculation (used to obtain main result) at $t/L = 0.1$ and 1 were obtained, but those at $t/L = 10$ were not obtained due to numerical instability. Therefore, the fractional calculation at $t/L = 0.1, 1, \text{ and } 10$ were also obtained. The comparison between the batch and fractional calculation was similar, so we discussed variation of model prediction at $t/L = 0.1, 1, \text{ and } 10$ using the results from both batch and fractional calculation. In the Nankai subduction zone, the model prediction $t/L = 10$ shows the occurrence of talc + carbonate at MWC and carbonate (magnesite) + quartz at the greater depth. These points are described at line 319–325 and Supplementary Discussion 2.

Moreover, in the calculated results were discussed based on the geologically constrained timescale. For example, the tectonic constraints of the Nankai subduction zone suggest that the subduction of the Philippine Sea Plate (re)initiated

at 17 Ma. Based on this, we assumed that time scale of fluid infiltration of 1–10 Myrs
Therefore, our model results represent

- After 1 Myr of fluid infiltration, our prediction suggests that a 1-m-thick layer at the base of the forearc mantle was composed of talc-poor serpentinite around the MWC and talc + carbonate rocks at greater depth (based on the results from $t/L = 1 \text{ Myr m}^{-1}$; Figs. 2g, h and 3a-c).
- After 10 Myr of fluid infiltration, our prediction suggests that the 1 m thick basement layer was transformed to talc + carbonate at 35-50 km to quartz + magnesite assemblage at 50-74 km (based on the results from $t/L = 10 \text{ Myr m}^{-1}$; Supplementary Discussion 2; Fig. 3d, e), but a 10 m thick basement layer composed of serpentinite around MWC and talc + carbonate rocks at greater depth was expected to occur simultaneously when continuous fluid infiltration occurs beyond the 1 m thick layer (based on the results from $t/L = 1 \text{ Myr m}^{-1}$; Figs. 2g, h and 3d, e).

We added these discussions (on lines 141–144 and 164 –173) and Figure 3b-e in the revised manuscript.

Although the talc-poor assemblage (carbonate + quartz) was predicted at the greater depth, we emphasize that the rheological contrast between MWC and the greater depth is still valid because a thicker layer rich in talc + carbonate rock was also predicted. This point is discussed in line 319–325.

Lines 148-152: Why not show these figures in the main text?

This is because (i) the journal has limited space (6000 words limit for the main text), and (ii) the calculation described here is the parameter study that is not directly relevant to the main topic of the manuscript (investigates the site-specific process involving the composition of subducting sediments and the thermal structure of each subduction zone).

Lines 153-155: I think you need a reference here to a study that demonstrates the abundance of carbonates being subducted in these subduction zones.

As suggested, we have added a reference (Clift 2017) that summarizes the global variation in total carbon, organic carbon, and inorganic carbon concentrations in subducting sediments. Line 254.

References

Clift, P. D. (2017). A revised budget for Cenozoic sedimentary carbon subduction. *Reviews of Geophysics*, 55(1), 97-125.
<https://doi.org/10.1002/2016RG000531>

Lines 155-157: I was a co-author on the study cited here. I'm not sure we *suggested* that talc was formed by Si metasomatism, so much as we explored whether Si metasomatism could produce rheologically significant amounts of talc at the Guerrero plate interface. Regardless, your point based on your findings in this study are valuable.

First of all, we appreciate your respective papers on whether Si metasomatism can produce rheologically significant amounts of talc at the Guerrero. Your work highly

motivated our study.

In the revised manuscript, we carefully rewrote the sentence, as follows (line 250); “Previous thermodynamic modeling of the Mexico subduction zone using a carbon-free system²⁶ has estimated the amounts of talc formed by Si metasomatism (reaction (1)). Our calculations suggest that the amount of talc predicted in this system is greater if subducted carbon in the Mexico subduction zone (carbonaceous materials and carbonate minerals)³² is considered.”.

Lines 212-214: is the talc disappearance thermally controlled by an imposed decoupling depth in the thermal model....and if so, would that not make your logic here circular?

After careful consideration, we believe the logic is correct.

The MDD is determined by thermal models that use data on subsurface heat flow and slab shape. Therefore, MDD is inferred independently of mineral stability, and the thermal model result can be combined with phase stability.

The proposed relationship between talc disappearance and MDD is based on the previous study (Peacock and Wang 2021), which uses closed-system phase equilibria. Here, we propose that talc stability under the closed-system calculation (Peacock and Wang 2021) is similar to our system calculations. In the revised manuscript, sentences were rewritten to emphasize that previous talc stability (closed system) is similar to our model (open system) (line 285–298).

References

Peacock, S. M. & Wang, K. On the Stability of Talc in Subduction Zones: A Possible Control on the Maximum Depth of Decoupling Between the Subducting Plate and Mantle Wedge. *Geophys. Res. Lett.* 48, 1–8 (2021).

I hope my comments and suggestions here are helpful.

We thank reviewer #2. The comments and suggestions are constructive and helpful.

Reviewer #3

The authors investigate the influence of mantle wedge fluid transport on spatially variable subduction zones. For this purpose, they use Perple_X, which is a commonly used thermodynamic modeling software. The authors specifically look at the subduction zones in Northeast and Southeast Japan, which have a different thermal structure and chemical compositions according to the authors. The authors conclude that the infiltration of carbon-bearing fluids causes heterogeneities in the mantle wedge, which could have an effect on the mechanical properties of the subduction zone. The paper is interesting, generally thorough, and well explained, with an important exception for the methods as stated below, which requires clarification. I am not able to fully review the geochemistry of the paper, since that is outside my field of expertise.

We are grateful for the reviewer's high evaluation of the paper and for the comments and suggestions that helped to improve the manuscript.

In the methods sections 2 and 3 (lines 492 to 529), two equations are stated to be used. The authors explain what the individual elements of the equation do and how it is used, but it is not explained where these equations come from and what their assumptions are. This needs to be addressed.

This equation is based on the mass balance. We clarified this point at line 377 and 401.

In the equation (3), we assumed that the thickness of sediment and the speed of subduction are constant. In the equation (4), we assumed that all fluid released from the subducting metasediments were infiltrated into the forearc mantle. In the revised manuscript, we mentioned these points at line 385 and 407, respectively.

In the code availability section, MinVel is not mentioned, while it is used.

We mentioned the availability of the MinVel software in the Code Availability section.

It is not clear to me whether the equations in the method section are computed by hand, or are part of a workflow of scripts and/or code. If code is used, it should be mentioned here.

Some parts of the calculations were done by hand. However, where appropriate, we developed scripts to automatically run Perple_X and process the results to increase the efficiency of the calculations. This is noted in line 422 and in the Data Availability section.

It would also be useful to be more clear about the actual workflow in the methods section.

In the revised manuscript, we added the sentences to explain the overview of the actual workflow (line 339). We hope that these revisions will improve the readability of the manuscript.

REVIEWERS' COMMENTS

Reviewer #1 (Remarks to the Author):

Oyanagi & Okamoto present a revised manuscript addressing C cycling and talc formation at forearc depths for two contrasting subduction zone P-T paths.

The authors have greatly increased the clarity of their methods and have made substantial edits that have improved the manuscript. My additional comments are very minor in nature. This manuscript will make an excellent contribution to this Journal.

Minor Comments:

Throughout: the term site-specific is used throughout the manuscript (e.g., L70, 72). I recommend using a different term, perhaps "margin-specific". The reason being that site-specific has a very specific meaning in petrology and mineralogy, referring to element exchange within the crystal lattice, and caught me off guard when I read it in the text.

L81: add "of"  representative "of" cold and . . .

L160-173: review and edit for clarity. This paragraph reports results for 1 Myr of infiltration with $t/L = 1$, 10 Myr of infiltration for $t/L = 10$, and 10 Myr of infiltration for $t/L = 1$? I suggest rearranging so that the discussion focuses first on $t/L = 1$ for both 1 Myr and 10 Myr of modeled flow, and then discuss how that is different with a slower $t/L = 10$.

Reviewer #2 (Remarks to the Author):

The authors have done a commendable job with their revisions in response to the reviews. I have very minor comments and suggestions for improvement that the authors might want to consider prior to publication.

1. The results and interpretations of the modelling are predicated on the availability of serpentine in the forearc mantle wedge to react to form talc + carbonate. The amount of serpentine available to react may dictate how much talc and carbonate can form, thus affecting the rheology of the plate interface. It is worth mentioning the results of a recently published study (Epstein et al., 2024, AGU Advances) which shows that different evolutionary stages of subduction (e.g. thermal regimes) result in different extents of serpentinization of the forearc mantle wedge. It may be useful to mention geologic evidence that the Nankai subduction zone is in a state of its evolution such that there is abundant serpentine in the forearc mantle wedge.

2. In line 63, I would argue against the claim that this is "unknown" - see the Epstein et al. (2024) paper mentioned above. Though that paper presents modelling (and not empirical observations), it at least gives quantitative modelling results to base our knowledge of the mineralogical variability of the mantle

wedge.

3. Line 81: "representative of"

4. Line 85-87: It might be useful to indicate the geothermal gradient in °C/GPa in parentheses, rather than leaving it up to the readers to calculate it themselves.

5. Line 210: "State" is a vague term - what do you mean by this?

6. Line 232-233: It might be helpful to preface this sentence with something like "We test whether our results for the Nankai subduction zone can be generalized more broadly to warm subduction zones by applying our methodology to the Cascadia, Mexico, and Colombia subduction zones."

7. Line 247: Insert "The" at the start of the sentence.

Reviewer #1 (Remarks to the Author):

Oyanagi & Okamoto present a revised manuscript addressing C cycling and talc formation at forearc depths for two contrasting subduction zone P-T paths. The authors have greatly increased the clarity of their methods and have made substantial edits that have improved the manuscript. My additional comments are very minor in nature. This manuscript will make an excellent contribution to this Journal.

We are very grateful to the reviewer (Dr. Gabe S. Epstein) for the constructive suggestions that helped us significantly improve the manuscript.

Minor Comments:

Throughout: the term site-specific is used throughout the manuscript (e.g., L70, 72). I recommend using a different term, perhaps "margin-specific". The reason being that site-specific has a very specific meaning in petrology and mineralogy, referring to element exchange within the crystal lattice, and caught me off guard when I read it in the text.

We agree. We removed the term "site-specific" throughout manuscript, and express it more clearly. For example, modified sentence (at line 78) was "Moreover, we considered the composition of the subducting carbon-bearing sediments that differ by each subduction zone".

L81: add "of"  representative "of" cold and . . .

We modified so.

L160-173: review and edit for clarity. This paragraph reports results for 1 Myr of infiltration with $t/L = 1$, 10 Myr of infiltration for $t/L = 10$, and 10 Myr of infiltration for $t/L = 1$? I suggest rearranging so that the discussion focuses first on $t/L = 1$ for both 1 Myr and 10 Myr of modeled flow, and then discuss how that is different with a slower $t/L = 10$.

As suggested, we arranged the discussion to focus first on $t/L = 1$ and then on $t/L = 10$. Line 175 – 181.

Reviewer #2 (Remarks to the Author):

The authors have done a commendable job with their revisions in response to the reviews. I have very minor comments and suggestions for improvement that the authors might want to consider prior to publication.

We are very grateful to the reviewer for the constructive suggestions that helped us significantly improve the manuscript.

1. The results and interpretations of the modelling are predicated on the availability of serpentine in the forearc mantle wedge to react to form talc + carbonate. The amount of

serpentine available to react may dictate how much talc and carbonate can form, thus affecting the rheology of the plate interface. It is worth mentioning the results of a recently published study (Epstein et al., 2024, AGU Advances) which shows that different evolutionary stages of subduction (e.g. thermal regimes) result in different extents of serpentinization of the forearc mantle wedge. It may be useful to mention geologic evidence that the Nankai subduction zone is in a state of its evolution such that there is abundant serpentine in the forearc mantle wedge.

We added Epstein et al. (2024) to the reference. Moreover, we added following sentences;

(Line 51) “Fluids derived from the subducted sediments are regarded as one of dominant fluid source of fluid-rock interactions in the forearc mantle throughout infancy to maturity of the subduction zone²⁵.”

(Line 65) “A coupled time-evolving subduction zone thermal model and phase equilibria show that temperature evolution from subduction infancy to maturity results in different degrees of serpentinization of the forearc mantle wedge²⁵”

2. In line 63, I would argue against the claim that this is "unknown" - see the Epstein et al. (2024) paper mentioned above. Though that paper presents modelling (and not empirical observations), it at least gives quantitative modelling results to base our knowledge of the mineralogical variability of the mantle wedge.

We agree. The sentence was weakened to read (Line 68);

“Understanding on the nature of depth-related variations in the mineral assemblage of the forearc mantle and its dependence on the thermal structure of the subduction zone is still limited, especially with respect to the interaction with multi-component fluids.”

We added Epstein et al. (2024) to the reference.

3. Line 81: "representative of"

We modified so.

4. Line 85-87: It might be useful to indicate the geothermal gradient in °C/GPa in parentheses, rather than leaving it up to the readers to calculate it themselves.

We indicated geothermal gradients (line 94)

5. Line 210: "State" is a vague term - what do you mean by this?

We clarified it as “dominant form of carbon (i.e., organic or inorganic carbon)” (Line 218)

6. Line 232-233: It might be helpful to preface this sentence with something like "We test whether our results for the Nankai subduction zone can be generalized more broadly to

warm subduction zones by applying our methodology to the Cascadia, Mexico, and Colombia subduction zones."

We appreciate this. We modified so.

7. Line 247: Insert "The" at the start of the sentence.

We modified so.